# Spinal Reflex Excitability of Lower Leg Muscles Following Acute Lateral Ankle Sprain: Bilateral Inhibition of Soleus Spinal Reflex Excitability

**DOI:** 10.3390/healthcare10071171

**Published:** 2022-06-23

**Authors:** Joo-Sung Kim, Kyung-Min Kim, Eunwook Chang, Hyun Chul Jung, Jung-Min Lee, Alan R. Needle

**Affiliations:** 1Department of Kinesiology and Sport Sciences, University of Miami, Coral Gables, FL 33146, USA; jjk147@miami.edu (J.-S.K.); km.kim@g.skku.edu (K.-M.K.); 2Department of Sport Science, Sungkyunkwan University, Suwon-si 16419, Korea; 3Department of Kinesiology, Inha University, Incheon 22212, Korea; change@inha.ac.kr; 4Department of Coaching, Kyung Hee University-Global Campus, Yongin-si 17014, Korea; jhc@khu.ac.kr; 5Department of Physical Education, Kyung Hee University-Global Campus, Yongin-si 17014, Korea; jungminlee@khu.ac.kr; 6Sports Science Research Center, Kyung Hee University-Global Campus, Yongin-si 17014, Korea; 7Department of Public Health & Exercise Science, Appalachian State University, Boone, NC 28608, USA; 8Department of Rehabilitation Science, Appalachian State University, Boone, NC 28608, USA

**Keywords:** ankle injuries, hoffmann reflex, arthrogenic muscle inhibition, neuroplasticity

## Abstract

Neural changes in the ankle stabilizing muscles following ankle sprains are thought to be one contributing factor to persistent ankle dysfunction. However, empirical evidence is limited. Therefore, we aimed to examine spinal reflex excitability of lower leg muscles following acute ankle sprains (AAS). We performed a case-control study with 2 groups consisting of 30 young adults with AAS and 30 aged-matched uninjured controls. Hoffmann reflex (H-reflex) testing was performed to estimate spinal reflex excitability of lower leg muscles: soleus, fibularis longus (FL), tibialis anterior (TA). Maximal H-reflex (H_max_) and motor responses (M_max_) were determined by delivering a series of electrical stimuli at the sciatic nerve. H_max_/M_max_ ratios were calculated to represent normalized spinal reflex excitability. Separate group-by-limb analyses of variance (ANOVA) with repeated measures found there were no significant interactions for any of the muscles (SL: *F*_1,56_ = 0.95, *p* = 0.33, FL: *F*_1,51_ = 0.65, *p* = 0.42, TA: *F*_1,51_ = 1.87, *p* = 0.18), but there was a significant main effect of group in the soleus (*F*_1,56_ = 6.56, *p* = 0.013), indicating the H_max_/M_max_ ratio of soleus in the AAS group was significantly lower bilaterally (AAS = 0.56 ± 0.19, control = 0.68 ± 0.17, *p* = 0.013), with no significant group differences in the other muscles (FL: *F*_1,51_ = 0.26, *p* = 0.61, TA: *F*_1,51_ = 0.93, *p* = 0.34). The bilateral inhibition of the soleus spinal reflex excitability following AAS may be significant in that it may explain bilateral sensorimotor deficits (postural control deficits) following unilateral injury, and provide insights into additional therapies aimed at the neural change.

## 1. Introduction

Ankle sprains are one of the most prevalent musculoskeletal injuries in general and physically active populations, with an estimate of 2 million ankle sprains occurring each year in the United States [1,2]. The lateral ligament complex of the ankle is composed of the anterior talofibular ligament (ATFL), calcaneofibular ligament (CFL) and posterior talofibular ligament (PTFL) [3], and these lateral ankle stabilizing ligaments are commonly injured by sudden rapid inversion and internal rotation of the foot and ankle complex [1,4]. The overall incidence rate for ankle sprains in soldiers was reported to be 45.14 per 1000 persons-years [5], while one-third of basketball players were affected by ankle sprains each season, accounting for a significant loss in playing time [2]. Due to the commonality of the injury in both general and athletic populations [1,2,5], the ankle sprain is often viewed as an innocuous injury. However, reinjury rates are high after initial acute ankle sprain (AAS) [2,6,7] and among those who sustained AAS, a large proportion of them (up to 70%) suffer from residual problems (for example, chronic pain and swelling, ankle dysfunction, disability, and lower health-related quality of life) for months to years after the injury. These lingering problems following AAS are thought to be due to mechanical and functional (sensorimotor) insufficiencies [8,9,10]. However, a collective body of evidence has found that sensorimotor insufficiencies are related to ankle dysfunction and disability [11]. Emerging evidence suggests there may be a neural alteration following AAS, which could contribute to residual symptoms and a chronic pathological condition, known as chronic ankle instability (CAI) [8].

A major debilitating condition that is often persistent following acute joint injuries is an inability to fully activate uninjured muscles surrounding the injured joint [12,13,14,15]. This ongoing neural inhibition has been clinically termed arthrogenic muscle inhibition (AMI) [16,17]. AMI is believed to be a contributing factor to various residual problems, such as profound muscle weakness [14,18], postural control deficits [19,20], abnormal movement patterns [21], and self-reported disability [22], which hinders effective rehabilitation across a range of musculoskeletal pathologies [17,23]. Although the underlying mechanism of AMI following the initial injury remains elusive, consistent literature utilizing Hoffmann reflex (H-reflex) methods suggests that AMI may be caused by alterations in spinal reflex pathways [13,14,17,24]. A decreased level of H-reflex following joint injury indicates the inability of muscles to be fully & rapidly activated [16,17,25]. Acute symptoms after the joint trauma, such as joint effusion [15,17,26] are thought to trigger the onset of AMI. For instance, a study of experimental ankle joint effusion reported immediate alterations of H-reflex amplitudes in ankle stabilizing muscles [15]. Similarly, knee joint effusion models demonstrated diminished H-reflex as well as decreased joint function [27,28]. Patients who underwent anterior cruciate ligament reconstruction (ACLR) showed a significant decrease in quadriceps H-reflex along with a marked reduction in quadriceps strength in the acute stage of injury [14]. Based on previous findings [14,15,27], there seems to be some consistency with respect to early changes in spinal reflex excitability of corresponding muscles following the initial joint injuries. However, these results have been found with either experimental effusion models or with a different population (knee joint injury) than AAS patients.

Research regarding neural excitability following AAS is very limited in comparison with CAI research: there are only two AAS studies [12,13] found in the literature compared with dozens of studies on CAI offering conclusive evidence of altered neural excitability in CAI patients that are linked to their sensorimotor deficits [18]. With respect to studies on AAS, Hall et al. [12] first examined spinal reflex excitability of ankle muscles following AAS and determined that AAS did not significantly change the spinal reflex excitability. However, the follow-up study by Klykken et al. [13] conversely reported that there were significant changes in the spinal reflex excitability of ankle muscles. These mixed results may be due to small sample sizes and different techniques to quantify spinal reflex excitability, and inconsistent criteria to determine the acute stage of an ankle sprain. With these factors significantly affecting previous findings, further studies are warranted to clarify these mixed results. Thus, the purpose of the current study was to examine spinal reflex excitability following AAS, with a larger sample size, standardized technique to quantify the spinal reflex excitability, and recommended criteria to determine the acute stage of an ankle sprain. Examining potential changes in neural excitability following AAS is critical to the current understanding of the underlying neurophysiological mechanism of sensorimotor dysfunction commonly seen in AAS patients. Further, it may influence the current paradigm of rehabilitation for AAS as there is emerging evidence that neural changes following joint injuries seem to be resistant to the current rehabilitation programs consisting of active exercises, requiring additional therapies aimed at the neural alterations [29,30]. We hypothesized that the spinal reflex excitability of ankle muscles significantly changes following AAS, and the neural changes would be associated with acute symptoms such as pain, ankle swelling, and self-reported dysfunction.

## 2. Materials and Methods

We employed a case-control study with two independent variables: group (AAS, healthy control) and limb (injured, uninjured). The primary outcomes were spinal reflex excitability of soleus, fibularis longus, and tibialis anterior, as assessed by normalizing maximal H-reflex (H_max_) to maximal motor response (M_max_) to calculate H_max_/M_max_ ratios. The secondary outcomes were pain, ankle swelling, and self-reported dysfunction.

### 2.1. Participants

A total of 30 patients with AAS and 30 healthy controls, who were matched for sex, age, weight, and height, participated in this study. Table 1 shows participants’ demographics. Inclusion criteria for the AAS group included a lateral ankle sprain affecting at least one of the lateral ankle ligaments such as anterior talofibular ligament (ATFL), calcaneofibular ligament (CFL), and posterior talofibular ligament (PTFL) [1,13]. The AAS group was included if they presented at least minimal pain/tenderness, swelling, and loss of function within the past three days (72 h) prior to participation [13]. Patients with AAS were excluded if they had current foot and ankle injuries other than the lateral ankle sprain (for example, fractures, high and medial ankle sprains), neurological injuries or diseases, cardiovascular diseases, cancer, or severe infection. Individuals with hypersensitivity to electrical stimulation or self-reported pregnancy were also excluded. Participants in the healthy control group were free of a history of ankle injury to either limb and did not report any lower extremity injuries within the previous six months. A licensed, certified athletic trainer assessed the current condition of injured ankles by using a standardized ankle injury evaluation form [13,31]. Ankle girths of both involved and uninvolved ankles were measured using the figure-of-eight method, and the ankle girth difference (involved girth-uninvolved girth, cm) was used to determine ankle swelling [12,31]. A current subjective pain level was assessed using a visual analog scale (VAS) [13]. Self-reported function was quantified by the Foot and Ankle Ability Measure (FAAM) activities of daily living (ADL) and sports subscales (Sport) [32]. The Institutional Review Board approved the study (IRB ID CON2013V4823). All of the participants provided informed written consent before they begin any study procedures.

### 2.2. Spinal Reflex Excitability

All of the participants in both groups (AAS and healthy control) underwent bilateral testing of spinal reflex excitability in the randomized order of test limb. An “injured-matched” limb of control participants was the same side of AAS patient’s injured limb. H-reflex tests were performed after a warm-up in which participants were asked to walk on a treadmill for 10 min at a self-selected pace. Investigators thoroughly prepared the skin by shaving and removing debris on the surface of electrodes. All of the participants were positioned prone on the massage table, and a padded plinth was placed under both legs to keep knee and ankle joint positions at a slight knee flexion and neutral positions, respectively. Throughout the experiment, participants listened to quiet relaxing music and stared at the picture of nature placed on the floor, which controlled their eye movements [33,34].

We placed a pair of pre-gelled Ag-AgCI 10-mm-adhesive/disposable surface electromyography (EMG) electrodes on the muscle belly of the soleus, fibularis longus, and tibialis anterior. Specifically, EMG electrodes for soleus muscle were placed around 2–3 cm distal in the midline of two heads of the gastrocnemius. For the fibularis longus, electrodes were positioned 2–3 cm distal to the head of the fibula. The electrodes for the tibialis anterior were placed at the mid-point at the muscle belly. For all of the pairs of electrodes, the inter-electrode distance was minimal (about 1.75 cm) in order to prevent any crosstalk from adjacent muscles [13,16,33,34]. A reference electrode was placed on the bony surface of the ankle at the medial malleolus. We inspected the quality and proper placement of EMG signals by monitoring resting EMG activities (0uV) during a completely relaxed position and active EMG activity by performing manual resistance testing of ankle plantarflexion, eversion, and dorsiflexion for soleus, fibularis longus, and tibialis anterior, respectively.

We utilized a data acquisition system (MP150; BIOPAC System, Inc., Goleta, CA, USA) to record reflex and motor responses during H-reflex testing. The sampling rate was set at 2000 Hz with a gain of 1000 [13,16,33,34]. The stimulator module (STMISOC; BIOPAC System, Inc., Goleta, CA, USA) was used to generate electrical stimuli for eliciting reflex responses, with a maximum stimulus output of 200 V. A 2 mm shield disk stimulating electrode (EL254Sl; BIOPAC System, Inc., Goleta, CA, USA) was also used to deliver electrical stimuli into the targeted nerve, with a 7 cm carbon-impregnated dispersive pad. The stimulating electrode was placed in the popliteal fossa, and the dispersive pad was put in the anterior suprapatellar region to make a complete electrical circuit with the stimulating electrode. Each measurement was recorded for an event window of 300 ms. We stimulated the sciatic nerve for simultaneously eliciting H-reflex responses from all 3 muscles. The stimulation occurred by a 1-ms square-wave electrical stimulus. A series of stimulations continued until maximal H-reflex (H_max_, 50–100 ms after the pulse) and motor responses (M_max_ 10–50 ms after the pulse) were found for all three leg muscles. Once H_max_ and M_max_ were identified, additional five stimuli were given at each H_max_ and M_max_ recording point, and these values were recorded. Five peak-to-peak values for H_max_ were averaged and normalized to averaged M_max_ to calculate the H_max_/M_max_ ratio, representing spinal reflex excitability. The soleus, fibularis longus, and tibialis anterior muscles were selected due to their functional role in ankle stabilization [20,35].

### 2.3. Statistical Analysis

An a priori power analysis was conducted in G*Power software to estimate a sample size required for the current study by using data of a previous study, which reported H_max_/M_max_ ratios in patients with and without AAS [13]. Based on the analysis, at least 15 participants would be needed per group to determine a significant difference with the α level of 0.05 and the statistical power (1 − β) of 0.80. Given the high variability in H-reflex measurements, we doubled the minimal sample size to 30 participants per group to provide conclusive evidence. We performed separate independent *t*-tests to compare demographic and secondary outcomes between groups. We also conducted separate 2 × 2 (group × limb) analyses of variance (ANOVA) with repeated measures to determine significant differences between groups (AAS and Control) and limbs (injured and uninjured) for H_max_/M_max_ ratios of soleus, fibularis longus, and tibialis anterior muscles. We calculated Cohen’s *d* effect sizes and its associated 95% confidence intervals (CI) to determine the magnitude of the group differences. The magnitude of effect sizes were interpreted as small (0.2 to 0.5), moderate (0.5 to 0.8), and strong (>0.8) [36]. After finding a group difference, we computed Pearson correlation coefficients (*r*) to examine the relationship of neural changes with acute symptoms such as pain (VAS scores), ankle swelling (ankle girth difference), and self-reported dysfunction (FAAM-ADL and FAAM-Sport scores). The correlation coefficients (*r*) were interpreted as following: weak (0 to 0.4), moderate (0.4 to 0.7), and strong (0.7 to 1.0). We used SPSS Statistics v25.0 (Armonk, NY, USA, IBM Corp), with the alpha level set a-prior at *p* < 0.05.

## 3. Results

### 3.1. Participant Demographics

Table 1 shows the descriptive data of the participant demographics. There were no significant differences in any demographic characteristics between the AAS and the control group, *p* > 0.05. However, the AAS group showed the presence of acute symptoms such as pain (VAS scores), ankle swelling (ankle girth difference), and self-reported dysfunction (FAAM-ADL and FAAM-Sport scores), *p* < 0.001.

### 3.2. Spinal Reflex Excitability

Table 2 shows descriptive statistics of H_max_/M_max_ ratios and Cohen’s d effect size estimates. For the soleus, there was no significant group-by-limb interaction (*F*_1,56_ = 0.952, *p* = 0.333). However, there was a significant group effect for the soleus indicating that the AAS group had a significantly lower soleus H_max_/M_max_ ratio than the control group regardless of limb, *F*_1,56_ = 6.56, *p* = 0.013. Effect sizes suggest this bilateral inhibition of spinal reflex excitabiliy was moderate: d = −0.65, 95% CI = −1.17 to −0.11. For the fibularis longus, there was neither significant group-by-limb interaction (*F*_1,51_ = 0.649, *p* = 0.424) nor group main effect (*F*_1,51_ = 0.264, *p* = 0.610). Similarly, there was neither group by limb interaction (*F*_1,51_ = 1.868, *p* = 0.178) nor group main effect (*F*_1,51_ = 0.928, *p* = 0.340) for the tibialis anterior. 

After finding the group difference in the soleus H_max_/M_max_, we examined the relationships between the neural changes with acute symptoms but did not find any significant relationships (Table 3).

## 4. Discussion

We found that patients with AAS had decreased spinal reflex excitability of the soleus muscle regardless of limb, indicating bilateral inhibition following unilateral injury. However, this reduced spinal reflex excitability was not correlated with severity of any of the acute symptoms: pain, ankle swelling, and self-reported ankle dysfunction. These findings suggest that the bilateral neural change may not be specifically mediated by local factors such as pain and ankle swelling and was not linked to self-reported ankle dysfunction. In addition to the soleus, other ankle muscles were not found to change their spinal reflex excitabilities following AAS.

### 4.1. Spinal Reflex Excitability Changes

Altered spinal reflex excitability has been reported in patients with joint trauma (ankle injuries [12,13,16,37,38,39] or knee injuries [14,40]). The H-reflex has been used to assess changes in spinal reflex excitability that represent the capability of alpha motoneurons of a muscle in the spinal cord to be excited by an external stimulus [41,42]. The reduction in the soleus H_max_:M_max_ ratios found in the current study may reflect the presence of AMI after AAS. Our finding appears to be consistent with prior investigations [16,24,37] that demonstrate altered neural excitability in patients with a history of ankle sprains. For example, McVey et al. examined H_max_:M_max_ ratios of soleus in patients with CAI and they found soleus H:M ratios in the injured limb were significantly smaller than the uninjured ankle [16]. These results reflect changes in the same direction of our findings; however, McVey et al. reported only a unilateral disturbance rather than bilateral inhibition. This could suggest that acute changes to the soleus may be bilateral and reflect centrally mediated mechanisms; however, the uninvolved side returns to a normal amount of excitability once acute symptoms (for example, pain & swelling) subside and functional status improves. Bowker et al. compared soleus H_max_:M_max_ ratios of CAI patients not only with controls (no history of ankle sprains) but also with copers whose ankles are functionally sound despite a history of ankle sprain [37]. They found significantly lower soleus H_max_:M_max_ ratios in patients with CAI when compared with controls or copers while no difference in ankle-joint laxity was observed, suggesting the presence of AMI regardless of mechanical alteration of the ankle (increased laxity) [37]. A recent meta-analysis examined a total of ten studies (154 patients with CAI and 138 healthy controls) that assessed soleus spinal reflex excitability and showed compelling evidence of the lesser spinal reflex excitability of soleus in patients with a history of ankle sprains compared with those without a history of ankle sprains [24].

However, the current findings are somewhat paradoxical in comparison with prior studies with AAS [12,13] that reported neither changes nor inhibited soleus H_max_:M_max_ ratios. Hall et al. [12] was the first study investigating H-reflex measures of ankle muscles in patients with acute inversion ankle sprain a week after onset [12]. They did not find any difference in H-reflex amplitudes in injured ankles compared with uninjured ankles. A direct comparison to our finding is not applicable because this study only assessed side-to-side differences in H-reflexes within the AAS group and did not use a healthy control group. A later study by Klykken et al. [13] examined the H-reflexes of soleus, fibularis longus, and tibialis anterior between the AAS group and controls, reporting no between-groups difference in H_max_:M_max_ ratios of any ankle muscles [13]. Additionally, when they compared the injured limb with the contralateral limb within the AAS group, soleus H_max_:M_max_ ratios resulted in facilitatory response while tibialis anterior in inhibitory response [13], which is opposite to the inhibition of soleus in the current study. Perhaps, such different responses in H-reflex after AAS may be due to variations in effects of acute injury symptoms on spinal reflex excitability. Klykken et al. [13] found that the increased pain was significantly correlated with reduced tibialis anterior H_max_:M_max_ ratios, speculating reciprocal inhibition occurring to the tibialis anterior in the previous study partly due to the perception of pain, that might also be associated with a facilitatory spinal response of soleus. However, the current study did not confirm any links between neural changes and severity of acute symptoms. The lack of associations may be supported by a recent systematic review, concluding that the presence of clinical pain does not influence H-reflex responses [43]. With regards to the effects of ankle swelling on spinal reflex excitability, a previous investigation [15] using simulated ankle joint effusion model showed alterations in H-reflex measures of ankle stabilizing muscles. However, the effusion model is utilized in healthy subjects with no previous ankle injury, limiting the understanding of the effects of swelling produced by actual injury on spinal reflex excitability. Previous investigations [12,13] with AAS patients found neither change nor a relationship for the effects of swelling on spinal reflex excitability. In this regard, our current findings, along with previous studies [12,13,43], indicate that underlying neural mechanism associated with AAS symptom severity does not appear to be directly related to alterations in spinal reflex excitability.

### 4.2. Bilateral Inhibition

The current study is unique in that it is the first to report bilateral inhibition of the soleus in patients with AAS. This bilateral inhibition is theorized to be associated with centrally mediated changes following unilateral joint trauma that has been previously reported in neuromuscular [40,44,45,46,47,48] and postural control measures [49,50,51]. Previous neuromuscular studies [40,44] examined descending motor functions to joint stabilizing muscles following unilateral joint injuries such as CAI [44] and anterior cruciate ligament injury [40], finding bilateral corticomotor deficits as assessed by transcranial magnetic stimulation. This suggests that descending supraspinal pathways may also be involved in bilateral deficits after unilateral ankle sprain, which interferes with efficient motor signals to corresponding joint muscles. Indeed, postural control studies [49,50,51] have demonstrated these centrally mediated changes. Evans et al. [50] examined the single leg balance of injured and uninjured ankles in patients with unilateral AAS and demonstrated that center of pressure excursion velocity was significantly increased in both ankles one day after the injury, indicating bilateral deficits. Another study [51] with postural control measures found that even a simple stance such as bipedal stance was significantly impaired after AAS. Along with these prior studies [50,51], a meta-analysis [49] confirmed a bilateral impairment with strong evidence of a postural impairment in the involved and uninvolved limb after AAS. Maintaining standing posture requires the activation of postural muscles to modulate the changes in the body’s center of pressure (COP). Oscillations in postural sway in the anterior to posterior direction occur as part of a feedforward mechanism, in which bilateral actions of the soleus plays a significant role in controlling quiet standing [52]. The soleus muscle predominantly consists of type 1 slow-switch muscle fibers (80% with a range of 64 to 100%) which primarily contributes to a static type of muscle activity, largely regulated through segmental mechanisms [53]. Therefore, the soleus muscle constantly acts to adjust the COP to correspond to sway excursions during static balance [52]. In this regard, these bilateral postural control impairments are likely indicative of centrally mediated changes, which have been observed in functional measures after ankle injuries. Our data on bilateral inhibition of the soleus supports this notion and suggests that bilateral soleus inhibition may be a contributing factor to the common bilateral postural impairments occurring immediately after the AAS. Future studies should test this hypothesis by investigating the relationship between bilateral spinal reflex excitability inhibition and postural impairments in order to determine centrally mediated changes in patients with AAS.

### 4.3. Selective Inhibition

We did not see any change in fibularis longus and tibialis anterior H_max_:M_max_ ratios in patients with AAS compared with healthy controls, which is partially in agreement with those previously reported [13]. However, it is still unknown which factors lead to selective changes in soleus spinal reflex excitability without affecting other ankle muscles after AAS as there are only two studies with AAS patients on spinal reflex excitability, and both studies were performed with a small sample size. A possible explanation might be the unique physiological characteristics of the soleus muscle. For instance, the physiological cross-sectional area (PCSA) of the soleus is significantly larger than the fibularis longus and tibialis anterior. The mean PCSA of the soleus is seven times larger than the fibularis longus and tibialis anterior [53], which might be associated with a predominant muscle work during physical activity [54]. The soleus is also composed of a higher percentage of slow-twitch fibers (type 1 fiber) with slow-twitch motor units compared with other ankle muscles [53]. Patients with AAS tend to avoid moving their ankles (functional immobilization). In the first few days of AAS, an acute joint inflammatory response is high, such that it upregulates gene expression involved with protein degradation and the elevated oxidative stress, which could result in a change in the fiber typification from slow-twitch to fast-twitch without affecting muscle fiber cross-sectional area [55]. It has been reported that soleus H_max_:M_max_ ratios were reduced in people with a predominance of fast-twitch muscle fibers compared with people with the predominance of slow-twitch fibers [55,56,57]. We may infer that our patients with AAS could have an acute inflammatory process according to the presence of acute symptoms (Table 2) [58], raising the speculation of involvement of molecular alterations in the change in soleus H_max_:M_max_ ratios following AAS [58,59], yet further research is needed to understand the mechanism of the selective inhibition of soleus after AAS.

### 4.4. Implications

The present study has important clinical implications for clinicians dealing with AAS. Anecdotal evidence suggests that current treatment for AAS is still focused on addressing acute injury symptoms, and therefore when the symptoms seem alleviated, individuals with AAS tend to quickly participate in their sports activities without considering the effects of AAS on their internal neurophysiological changes. However, based on our data, bilateral neural inhibition of the soleus can occur unrelated to the level of symptomatic response of their acutely sprained ankle. The bilateral inhibition of the soleus might lead to balance problems, which are associated high risk of recurrent ankle sprains. Therefore, clinicians need to address this immediate neural alteration by utilizing alternative therapeutic modalities that can effectively improve inhibited neural excitability of the soleus. A previous study showed that the soleus H-reflex could be upregulated with an operant conditioning method [60]. Focal ankle joint cooling has been reported to have not only a neuromodulatory effect on the soleus but also an analgesic effect after joint injury [61,62]. However, there is still little research in this area. Investigating the efficacy of these modalities in improving soleus spinal reflex excitability following AAS would help develop an innovative treatment option for acutely sprained patients.

### 4.5. Limitations

Our study has limitations. H-reflex measurement was limited to the prone condition, which might prevent a direct interpretation of our findings to the functional deficits after AAS. Although previous findings [12,13,24,37,38] in the sports medicine literature have demonstrated a change in H-reflex assessed during a lying condition after ankle sprains, a direct assessment of H-reflex during functional conditions (for example, single leg balance) provides insight into how neural excitability changes while patients with AAS maintain balance. For example, previous observations [19,20] suggested task-specific neural adaptations of ankle muscles during different postural conditions in patients with CAI. However, it was difficult to perform H-reflex testing in a weight-bearing position because participants with AAS reported increased levels of symptoms that preclude weightbearing assessment. Another limitation is that our study is limited to a retrospective design which only allows us to examine neural changes after the injury occurs. Given that there was marked ankle swelling and pain, it may be reasonable to state that the spinal reflex excitability change observed in the current study was due to the acute ankle trauma rather than spinal alteration that existed before the injury. Additionally, because of the acute aspects of the injury, it was unfortunately not possible to mask the investigator to injury status as signs of injury were evident. Finally, the study findings are limited to young adults, and they may differ in other age groups.

## 5. Conclusions

The present study found in patients with AAS, bilateral inhibition of the spinal reflex excitability in the soleus muscle, but this neural change was not found in other ankle muscles: fibularis longus and tibialis anterior. The soleus inhibition was not associated with any of acute symptoms such as pain, ankle swelling, and self-reported ankle dysfunction. As bilateral sensorimotor deficits (such as bilateral impaired postural control) are commonly associated with AAS, future studies may serve to determine the degree to which our documented changes in spinal excitability to the soleus impact clinical function.

## Figures and Tables

**Table 1 healthcare-10-01171-t001:** Participant demographics of the acute ankle sprain (AAS) and control group (mean ± SD).

Group	AAS (*n* = 30)	Control (*n* = 30)	*p* Value
Sex (male/female)	17/13	17/13	N/A
Age (yrs)	22.1 ± 4.3	22.1 ± 2.1	0.878
Height (cm)	174.8 ± 9.3	173.6 ± 10.2	0.645
Weight (kg)	74.3 ± 11.4	71.3 ± 14.0	0.371
Ankle swelling (cm) ^a^	1.5 ± 1.1 *	0.06 ± 0.3	<0.001
VAS score for pain (cm) ^b^	3.6 ± 1.7 *	0	<0.001
FAAM-ADL (%) ^c^	60.4 ± 21.2 *	99.6 ± 0.8	<0.001
FAAM-Sport (%) ^d^	34.8 ± 23.4 *	99.9 ± 0.6	<0.001

AAS, acute ankle sprain; N/A, not applicable; VAS, visual analog scale; FAAM, foot and ankle ability measure; ADL, activities of daily living. ^a^ Quantified by the figure-of-eight method, subtracted the average of three measurements of the uninjured ankle girth from the average of the injured one. A higher value represents greater ankle swelling. ^b^ Measured using the distance (cm) from the left edge (0 cm) to the perceived pain intensity on a 10 cm horizontal line. A higher score indicates a greater pain. ^c^ Self-reported ankle function during daily activities was measured within the past 3 days since the acute ankle sprain. A score less than 90 represents ankle dysfunction, with a lower score being worse dysfunction. ^d^ Self-reported ankle function during sports activities was measured within the past 3 days since the acute ankle sprain. A score less than 80 represents ankle dysfunction, with a lower score being worse dysfunction. * Significantly different from the control group.

**Table 2 healthcare-10-01171-t002:** H_max_/M_max_ ratios of lower leg muscles.

Muscles	Side	AAS Group	Control Group	Group Effect Size ^a^
Soleus	Injured	0.58 ± 0.20	0.67 ± 0.15	−0.51(−1.03, 0.02)
	Uninjured	0.55 ± 0.20	0.68 ± 0.19	−0.67(−1.19, −0.13)
	Combined ^b^	0.56 ± 0.19 *	0.68 ± 0.17	−0.65(−1.17, −0.11)
Fibularis longus	Injured	0.21 ± 0.14	0.21 ± 0.11	0.00(−0.54, 0.54)
	Uninjured	0.21 ± 0.13	0.19 ± 0.11	0.17(−0.38, 0.70)
	Combined ^b^	0.21 ± 0.13	0.20 ± 0.11	0.08(−0.46, 0.62)
Tibialis anterior	Injured	0.16 ± 0.13	0.18 ± 0.11	−0.17(−0.70, 0.38)
	Uninjured	0.16 ± 0.10	0.21 ± 0.14	−0.41(−0.95, 0.14)
	Combined ^b^	0.16 ± 0.12	0.19 ± 0.13	−0.24(−0.78, 0.31)

AAS, acute ankle sprain. ^a^ Cohen’s *d* estimate of effect size was calculated between two groups using pooled standard deviation, along with its associated 95% confidence interval. Negative (−) values represent the lower H_max_/M_max_ ratios of the AAS group, compared with the control group. ^b^ Indicates the pooled H_max_/M_max_ ratio data from both the injured and uninjured sides. * Significantly lower H_max_/M_max_ ratio in the AAS group, compared with the control group.

**Table 3 healthcare-10-01171-t003:** Relationships between reduced spinal reflex excitability of soleus and acute symptoms.

	Soleus H_max_:M_max_ Ratios ^a^	
	*r*	*p*
Ankle swelling (cm) ^b^	−0.25	0.18
VAS score for pain (cm) ^c^	−0.12	0.53
FAAM-ADL (%) ^d^	0.21	0.28
FAAM-Sport (%) ^e^	0.21	0.27

^a^ H_max_:M_max_ ratios from the injured limb of the acute ankle sprain group. ^b^ Quantified by the figure-of-eight method, subtracted the average of three measurements of the uninjured ankle girth from the average of the injured one. A higher value represents greater ankle swelling. ^c^ Measured using the distance (cm) from the left edge (0 cm) to the perceived pain intensity on a 10 cm horizontal line. A higher score indicates a greater pain. ^d^ Self-reported ankle function during daily activities was measured within the past 3 days since the acute ankle sprain. A score less than 90 represents ankle dysfunction, with a lower score being worse dysfunction. ^e^ Self-reported ankle function during sports activities was measured within the past 3 days since the acute ankle sprain. A score less than 80 represents ankle dysfunction, with a lower score being worse dysfunction.

## Data Availability

Not applicable.

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
