# Peer review of "Spinal Reflex Excitability of Lower Leg Muscles Following Acute Lateral Ankle Sprain: Bilateral Inhibition of Soleus Spinal Reflex Excitability"

_healthcare, 2022, doi:10.3390/healthcare10071171_

Round 1
Reviewer 1 Report
TITLE
- P. 1, line 2. Given the content of the article, you should consider to modify the title. If your objective was to examine spinal reflex excitability of lower leg muscles following acute ankle sprains (AAS), specifying only Soleus Spinal Reflex in the tittle don´t reflect the complete content of the article.
ABSTRACT
- P. 1 line 24. Please, in the results section, could you add some numerical/statistical values?
- P. 1 line 35. In the discussion section, you can also focus on the clinical application.
INTRODUCTION
The introduction tries to states the rationale for the study, but maybe makes the scientific background too broad, causing the reader to get lost, it could be more adequate to be more direct in the object of the study.
- P.1, line 43. Please, could you actualize the reference 2, specifying other ligaments as
the lateral fibulotalocalcaneal ligament complex?
- P.2, line 46. It could be interesting to include information about the prevalence or socio-demographic data of these injuries.
METHODS
- P.3, line 108. Given the inclusion criteria, please, specify in the tittle: inversion ankle sprain incurs in any way.
- P.3, line 111. Could you reference the criteria of AAS selected for the study?
- P.3, line 119. I don´t know if a certified Athletic trainer is the better professional to assessed the current injured condition.
- P.3, line 120. Could you specify if the researcher was blinded?
- P.3, line 141. Probably, due to the differences between the muscle dimensions, it could be better to use smaller electrode sizes for fibularis longus or tibialis anterior.
- P.4, line 171. Please, could you reference the follow sentence?: “The soleus, fibularis longus, and tibialis anterior muscles were selected due to their functional role in ankle stabilization”.
- P.4, line 172. The statistical analysis section should include an estimation of the sample size, justifying the main differences with previous article.
RESULTS
- P.5, line 194. It could be interesting to include different age groups, not only young participants.
- P.6, line 225. Please, could you introduce a table or an illustration of the relationship of the soleus neural changes with the acute symptoms?
DISCUSSION
- P.6, line 240. I think the hold sentence between the line 240 and 244 should be in the introduction section.
- P.6, line 250. It is not clear why the results of Mcvey et al. are in line with your results, they showed a unilateral disturbance instead of a bilateral inhibition of soleus spinal reflex excitability, could you rewrite this section in order to do it more adequate?
- P.8, line 350. This speculation should be supported with any reference. The previous sentence also has no reference.
CONCLUSION
- P.9, line 390. I think it is better to rewrite the last sentence as future needed studies. The way that the sentence is expressed would be appropriated if any sensorimotor deficits as the impaired postural control had been assessed.
Author Response
Responses to Reviewers’ comments
We would like to thank the editor and reviewers for your thorough review and observations to improve this manuscript. We have addressed all the comments, which would help strengthen the quality of the manuscript.
|
REVIEWER 1 |
|
Title |
|
Comment 1: - P. 1, line 2. Given the content of the article, you should consider to modify the title. If your objective was to examine spinal reflex excitability of lower leg muscles following acute ankle sprains (AAS), specifying only Soleus Spinal Reflex in the tittle don´t reflect the complete content of the article.
Author Response: We modified the title as suggested: “Spinal Reflex Excitability of Lower Leg Muscles following Acute Lateral Ankle Sprain: Bilateral Inhibition of Soleus Spinal Reflex Excitability”
|
|
Abstract |
|
Comment 2: - P. 1 line 24. Please, in the results section, could you add some numerical/statistical values?
Author Response: We added all statistical results to the abstract as suggested: P.1 line 28-33
|
|
Comment 3: - P. 1 line 35. In the discussion section, you can also focus on the clinical application.
Author Response: With the word limit, we added a sentence (phrase) on the clinical application as suggested: P.1 line 33-35 |
|
Introduction |
|
Comment 4: The introduction tries to states the rationale for the study, but maybe makes the scientific background too broad, causing the reader to get lost, it could be more adequate to be more direct in the object of the study.
Author Response: We thank the reviewer for this comment. We have condensed and/or removed some of the broad context in this draft of the manuscript to more directly place the study in its correct context. |
|
Comment 5: - P.1, line 43. Please, could you actualize the reference 2, specifying other ligaments as the lateral fibulotalocalcaneal ligament complex?
Author Response: As suggested, the lateral ankle ligaments were specified, and additional information was added: P.1 line 41-43
|
|
Comment 6: - P.2, line 46. It could be interesting to include information about the prevalence or socio-demographic data of these injuries.
Author Response: Suggested information about the prevalence was added: P.1 line 44-46
|
|
Material and Methods |
|
Comment 7: - P.3, line 108. Given the inclusion criteria, please, specify in the tittle: inversion ankle sprain incurs in any way.
Author Response: We specified the type of ankle sprain as lateral ankle sprain in the title as suggested.
|
|
Comment 8: - P.3, line 111. Could you reference the criteria of AAS selected for the study?
Author Response: Relevant references were added: P.3 line 112-114
|
|
Comment 9: - P.3, line 119. I don´t know if a certified Athletic trainer is the better professional to assessed the current injured condition.
Author Response: Certified athletic trainers are highly qualified, multi-skilled health care professionals who render service or treatment, under the direction of or in collaboration with a physician. As a part of the health care team, services provided by athletic trainers include primary care, injury and illness prevention, wellness promotion and education, emergent care, examination and clinical diagnosis, therapeutic intervention and rehabilitation of injuries and medical conditions. Injury assessment is one of primary tasks athletic trainers do on their daily practice. So, they are well suited to assess the status of patients with acute injuries including acute lateral ankle sprains. Further, the National Athletic Trainers’ Association is the organization who puts out the guidelines on the current practices in prevention, assessment, and treatment for acute ankle sprains and chronic ankle instability.
|
|
Comment 10: - P.3, line 120. Could you specify if the researcher was blinded?
Author Response: The researcher who assessed acute symptoms was not (could not be) blinded. We have acknowledged this in the discussion as a study limitation. P.11, line 392-392.
|
|
Comment 11: - P.3, line 141. Probably, due to the differences between the muscle dimensions, it could be better to use smaller electrode sizes for fibularis longus or tibialis anterior.
Author Response: We appreciate he reviewer’s point; however, we followed protocols from a study that has demonstrated reliability with similar electrode sizes across the tibialis anterior and fibularis longus (Palmieri et al. Int J Neurosci 2002) and has been utilized in multiple studies (McVey et al. Foot Ankle Int 2005, Palmieri-Smith et al. Am J Sports Med 2009, Klykken et al. J Athl Training 2011, Kim et al. J Electromyogr Kinesiol. 2012, Kim et al. J Athl Training 2016). Keeping this same methodology in the present study aids in comparing our results wth those of previous studies, and therefore we decided to keep the same method including the electrode size and inter-electrode distance.
|
|
Comment 12: - P.4, line 171. Please, could you reference the follow sentence?: “The soleus, fibularis longus, and tibialis anterior muscles were selected due to their functional role in ankle stabilization”.
Author Response: Relevant references were added: P.5, line 169 |
|
Comment 13: - P.4, line 172. The statistical analysis section should include an estimation of the sample size, justifying the main differences with previous article.
Author Response: As suggested, we added our estimation of the sample size and rationale for the current sample size: P.5 line 171-175
|
|
Results |
|
Comment 14: - P.5, line 194. It could be interesting to include different age groups, not only young participants.
Author Response: It is the great point to include different age groups. However, we only tested young adults, which should be one of the study limitations. We acknowledge this limitation: P. 11 line 393-394
|
|
Comment 15: - P.6, line 225. Please, could you introduce a table or an illustration of the relationship of the soleus neural changes with the acute symptoms?
Author Response: As suggested, we created the table 3, showing the relationships between reduced spinal reflex excitability of soleus and acute symptoms: P.7 |
|
Discussion |
|
Comment 16: - P.6, line 240. I think the hold sentence between the line 240 and 244 should be in the introduction section.
Author Response: As suggested, we revised the sentences between the line 240 and 244 to appropriately place them into the introduction: P.2. line 67-69
|
|
Comment 17: - P.6, line 250. It is not clear why the results of Mcvey et al. are in line with your results, they showed a unilateral disturbance instead of a bilateral inhibition of soleus spinal reflex excitability, could you rewrite this section in order to do it more adequate?
Author Response: The reviewer brings up an excellent point. A key difference between these studies is that McVey et al. investigate individuals with chronic ankle instability whereas we studied acute ankle sprain. These results reflect the same direction of changes, but may offer insight into the transition from AAS to CAI. We have added language to reflect this change. P.8 line 258-265 |
|
Comment 18: - P.8, line 350. This speculation should be supported with any reference. The previous sentence also has no reference.
Author Response: We added relevant references: P.9 line 358
|
|
Conclusion |
|
Comment 19: - P.9, line 390. I think it is better to rewrite the last sentence as future needed studies. The way that the sentence is expressed would be appropriated if any sensorimotor deficits as the impaired postural control had been assessed.
Author Response: The final sentence has been amended in line with the reviewer’s recommendations. P.11 Line 399-402 |
Reviewer 2 Report
“Bilateral Inhibition of Soleus Spinal Reflex Excitability following Acute Ankle Sprain: Central Neurophysiological Changes following Unilateral Injury”
Overall strengths of the article:
This manuscript investigates the spinal reflex excitability of lower leg muscles following acute ankle sprains (AAS). The study examines spinal reflex excitability following AAS, with relatively larger sample size, utilized a standardized technique to quantify the spinal reflex excitability, and recommended criteria to determine the acute stage of an ankle sprain. They observed that patients with AAS had decreased spinal reflex excitability of the soleus muscle regardless of limb, indicating bilateral inhibition following unilateral injury. However, this reduced spinal reflex excitability was not correlated with the severity of any of the acute symptoms: pain, ankle swelling, and self-reported ankle dysfunction. A Well detailed and executed paper examining the spinal reflex excitability post-AAS. Overall, the methods are well detailed, the statistical analysis is rigorous, and the data well presented (except for maybe a few critiques, detailed below).
Specific comments on weaknesses:
Major Critical Comments:
1. A stimulus-response plot for H-reflex and M-wave would be helpful to understand the time course of events.
2. After AAS, what are the factors responsible for the changes in soleus spinal reflex excitability? And why is has no effect on the other ankle muscles?
3. H-reflex data were quired only in the prone condition, what could be the effect of other conditions on the current findings?
Minor points:
1. Minor formatting issue, page 1, lines 11 &19.
2. References need to be formatted carefully as in many years of publication are in bold and in others, it is not bold (e.g., ref. 30)
3. Line 534; Ref. 58; ‘doi’ is written twice, please remove one of them.
Author Response
Responses to Reviewers’ comments
We would like to thank the editor and reviewers for your thorough review and observations to improve this manuscript. We have addressed all the comments, which would help strengthen the quality of the manuscript.
|
REVIEWER 2 |
|
General comment |
|
“Bilateral Inhibition of Soleus Spinal Reflex Excitability following Acute Ankle Sprain: Central Neurophysiological Changes following Unilateral Injury” Overall strengths of the article:
This manuscript investigates the spinal reflex excitability of lower leg muscles following acute ankle sprains (AAS). The study examines spinal reflex excitability following AAS, with relatively larger sample size, utilized a standardized technique to quantify the spinal reflex excitability, and recommended criteria to determine the acute stage of an ankle sprain. They observed that patients with AAS had decreased spinal reflex excitability of the soleus muscle regardless of limb, indicating bilateral inhibition following unilateral injury. However, this reduced spinal reflex excitability was not correlated with the severity of any of the acute symptoms: pain, ankle swelling, and self-reported ankle dysfunction. A Well detailed and executed paper examining the spinal reflex excitability post-AAS. Overall, the methods are well detailed, the statistical analysis is rigorous, and the data well presented (except for maybe a few critiques, detailed below).
|
|
Specific comments on weakness: major critical comments |
|
Comment 1: A stimulus-response plot for H-reflex and M-wave would be helpful to understand the time course of events.
Author Response: It is the great idea to have a stimulus-response plot for H-reflex and M-wave, but it turned out to be difficult to build the recruitment curve consistently across individuals with AAS due to the varying levels of tolerance of discomfort associated with electrical stimuli eliciting either H-reflex and M-wave. So, we focused on finding maximal H-reflex and M-wave for the outcome of Hmax:Mmax ratio.
|
|
Comment 2: After AAS, what are the factors responsible for the changes in soleus spinal reflex excitability? And why is has no effect on the other ankle muscles?
Author Response: The reviewer brings up an excellent point. We had attempted to address this in section 4.3, “Selective Inhibition”. In the current version of the manuscript we have expanded this section to better address the reviewer’s comment, but our data leaves us unable to definitively explain this other than speculating towards the differences in soleus innervation that might affect this. |
|
Comment 3: H-reflex data were quired only in the prone condition, what could be the effect of other conditions on the current findings?
Author Response: The reviewer brings up another excellent point. In acute ankle sprain models, maintaining bipedal or unipedal balance for the duration of time it would take to assess the H-reflex would likely exacerbate patients’ symptoms; however, this is acknowledged within the limitations section. P.10 line 385-387 |
|
Specific comments on weakness: minor points |
|
Comment 5: Minor formatting issue, page 1, lines 11 &19.
Author Response: Formatting was corrected. |
|
Comment 6: References need to be formatted carefully as in many years of publication are in bold and in others, it is not bold (e.g., ref. 30)
Author Response: A list of references was thoroughly reviewed and correctly formatted.
|
|
Comment 3: Line 534; Ref. 58; ‘doi’ is written twice, please remove one of them.
Author Response: Duplicate ‘doi’ is removed as suggested.
|